# Real World Patient Eligibility for Second Line Lurbinectedin Based Treatment in Small Cell Lung Cancer: Understanding Epidemiology and Estimating Health Care Utilization

Rebekah Rittberg [1,2], Bonnie Leung [1], Zamzam Al-Hashami [3] and Cheryl Ho [1,*]

1. BC Cancer, Department of Medical Oncology, Vancouver, BC V5Z 4E6, Canada
2. CancerCare Manitoba, Section of Hematology/Oncology, Department of Internal Medicine, University of Manitoba, Winnipeg, MB R3E 0V9, Canada
3. Sultan Qaboos Comprehensive Cancer Care and Research Center, Muscat P.O. Box 566 P.C 123, Oman
* Correspondence: cho@bccancer.bc.ca; Tel.: +1-604-877-6000

**Abstract:** Background: In the ATLANTIS study, second-line lurbinectedin/doxorubicin did not improve overall survival (OS), however patients with a chemotherapy-free interval (CTFI) of $\geq$180 days had an improved progression free survival (PFS). The objective of this retrospective study was to identify the proportion of real-world small cell lung cancer (SCLC) patients who are suitable for lurbinectedin-based therapy based on these criteria. Methods: A retrospective study of all SCLC referred to BC Cancer between 2012 and 2017 was conducted. Patient demographics, staging, treatment, and survival data were collected retrospectively. Baseline characteristics were compared using descriptive statistics. OS was calculated using Kaplan–Meier curves. Statistically significant *p*-value was <0.05. Results: A total of 1048 patients were identified. Baseline characteristics: median age 68 years, 47% male, 61% current smoking status, 68% extensive disease. Best supportive care was received by 22%. First-line systemic therapy was platinum doublet for 71% of the population. Second-line systemic therapy was delivered to 22%. Of the 219 patients who received second-line systemic therapy after platinum doublet, 183 patients had a CTFI of $\geq$90 days and 107 patients had a CTFI of $\geq$180 days. Patients originally treated as limited stage disease, received platinum doublet as second line, received thoracic radiation (RT) or prophylactic cranial irradiation (PCI) were more likely to have a CTFI of $\geq$90 and $\geq$180 days. Conclusion: In our real-world SCLC population, only 21% of the SCLC population received second-line therapy after platinum doublet with 17% achieving CTFI of $\geq$90 days and 10% CTFI of $\geq$180 days. Based on this retrospective review, only a small fraction of platinum-treated patients would be preferentially offered lurbinectedin in the second-line setting.

**Keywords:** SCLC; small cell lung cancer; lurbinectedin; chemotherapy-free interval; second line; real world evidence

## 1. Introduction

Small cell lung cancer (SCLC) is a highly aggressive, rapidly progressing neuroendocrine tumor accounting for 15% of new lung cancer diagnoses [1]. Tobacco use accounts for the vast majority of new cases of SCLC with 98% of patients having a smoking history [1]. Due to the aggressive nature of SCLC, two thirds of patients present with incurable extensive stage (ES) disease with a median overall survival (OS) of just 10 months [2,3]. Unfortunately, even in the minority of patients who present with limited stage (LS) disease, treatment is rarely curable with only 16% of patients being alive at 5 years [4].

SCLC is highly responsive to first-line platinum doublet-based chemotherapy, with response rates of 60–70% [1,5]. Recent advances in therapy for patients with ES include the addition of immune checkpoint inhibitors, atezolizumab and durvalumab, respectively, to first-line platinum doublet. The combination therapy resulted in improved OS, representing the most significant advance in SCLC over the past two decades [6,7].

Despite the initial good response to first-line treatment, most patients relapse within 6 months with much lower response rates to second-line therapy [8–10]. If patients are platinum sensitive, defined as relapse greater than 90 days from last receiving a platinum agent, patients can be rechallenged [11]. Platinum refractory SCLC has a lower rate of disease response with options including cyclophosphamide, doxorubicin and vincristine (CAV) and single agent topoisomerase I inhibitors; topotecan and irinotecan [8–10].

In the second-line setting, multiple therapies have been explored including immunotherapy, antibody drug conjugates, VEGF inhibitors, PARP inhibitors and transcription inhibitors with mixed results [12]. Recent success has been seen with lurbinectedin, an oncogenic transcription inhibitor that prevents DNA damage repair and degrades transcribing RNA polymerase II [12–14]. Single agent lurbinectedin in the second-line treatment of ES SCLC, demonstrated an overall response rate (ORR) of 35%. In a single arm study, the median OS was 9.3 months in the entire population and 11.9 months in patients with a chemotherapy-free interval (CTFI) of ≥90 days [13]. Based on these phase II results, lurbinectedin received accelerated approval from the Food and Drug Administration in the United States of America.

Lurbinectedin has also been explored in combination with other agents [14]. In a phase I trial, doxorubicin and lurbinectedin were evaluated with a dose escalation design and were found to have an ORR of 58% with a median duration of response of 4.5 months (NCT01970540) [15]. This led to ATLANTIS, a phase III randomized control trial evaluating lurbinectedin and doxorubicin compared to topotecan or CAV in LS and ES SCLC patients who had received at least one prior platinum-based chemotherapy. The OS primary endpoint was not met. On subgroup analysis, median progression-free survival (PFS) was 8.2 versus 4.5 months in patients with CTFI of ≥180 days [16,17].

SCLC continues to have unmet systemic therapy needs. Here we undertook a retrospective study evaluating the epidemiology of a SCLC cohort over 8 years. This cohort was then used to evaluate what proportion or which ES SCLC subset would most benefit from lurbinectedin-based therapy. Based on the current published results of lurbinectedin, it is not clear what proportion or which ES SCLC subset would most benefit from this therapy. Using CTFI, we retrospectively evaluated SCLC patients to determine the potential uptake of lurbinectedin.

## 2. Methods

### 2.1. Population

BC Cancer is a provincial cancer agency serving a population of 5.2 million people through 6 cancer centers, allowing for centralized organization of cancer treatment. All referred lung cancer patients are registered in the Outcomes and Surveillance Integration System, which contains baseline patient demographic records, disease characteristics and patient outcomes.

A retrospective cohort study was conducted in British Columbia, Canada evaluating patients referred to BC Cancer with SCLC between 1 January 2010 and 31 December 2017. Patient demographics, Eastern Cooperative Oncology Group (ECOG) performance status (PS), staging, treatment and survival data were collected using the Outcomes and Surveillance Integration System, electronic medical records, billing administration database and provincial chemotherapy records. Radiation data were obtained through the electronic treatment records. Missing data were abstracted manually through chart review.

CTFI was defined as the time from last dose of first-line chemotherapy administered to the first dose of the subsequent chemotherapy regimen. OS was calculated from date of diagnosis to death.

### 2.2. Statistical Analysis

Descriptive statistics were produced using Statistical Package for the Social Software version 28. Statistical tests included Chi-square and Mann–Whitney U tests. OS was calculated using the Kaplan–Meier curves and compared using the log rank test. Patients

were censored at time of last known follow up. Statistically significant *p*-values was set at <0.05.

### 2.3. Ethics Statement

This study was conducted with the approval of the University of British Columbia/BC Cancer Research Ethics Board (H19-02381). A waiver of consent was granted to extract and analyze data for this retrospective review.

## 3. Results

Between 2010 and 2017, 1048 patients were diagnosed with SCLC. Baseline characteristics; median age 68 years and 47% male. Current smokers made up 61% of the population and former smokers made up 36% of the population. Baseline ECOG PS was 0–1 for 30%, 2 for 22%, 3–4 for 28% and unknown 20%. Stage included 32% (*n* = 333) LS disease and 69% (*n* = 715) had ES disease (Table 1).

**Table 1.** Baseline characteristics from full SCLC population and patients treated with first-line platinum doublet who received second-line systemic therapy based on chemotherapy-free interval.

| N (%) | Overall (*n* = 1048) | Chemotherapy-Free Interval after Platinum Doublet | | | | | |
|---|---|---|---|---|---|---|---|
| | | <90 Days (*n* = 36) | ≥90 Days (*n* = 183) | *p*-Value | <180 Days (*n* = 112) | ≥180 Days (*n* = 107) | *p*-Value |
| Age (median), years | 68 | 60 | 65 | 0.173 | 66 | 64 | 0.467 |
| Sex | | | | | | | |
| Male | 497 (47%) | 14 (39%) | 86 (47%) | | 52 (46%) | 48 (45%) | |
| Female | 551 (53%) | 22 (61%) | 97 (53%) | 0.24 | 60 (54%) | 59 (55%) | 0.461 |
| Stage at diagnosis | | | | | | | |
| Limited | 333 (32%) | 3 (8%) | 79 (43%) | | 29 (26%) | 53 (49%) | |
| Extensive | 715 (68%) | 33 (92%) | 104 (57%) | <0.001 | 83 (74%) | 54 (51%) | <0.001 |
| Smoking Status | | | | | | | |
| Never | 23 (2%) | 0 | 5 (3%) | | 0 | 5 (5%) | |
| Former | 376 (36%) | 14 (39%) | 57 (31%) | | 44 (39%) | 27 (25%) | |
| Active | 639 (61%) | 22 (61%) | 121 (66%) | | 68 (61%) | 75 (70%) | |
| Unknown | 10 (1%) | 0 | 0 | 0.437 | 0 | 0 | 0.511 |
| Smoking years (median) | 50 | 40 | 40 | 0.667 | 40 | 45 | 0.207 |
| ECOG PS | | | | | | | |
| 0–1 | 319 (30%) | 9 (25%) | 92 (50%) | | 48 (43%) | 53 (50%) | |
| 2 | 228 (22%) | 9 (25%) | 41 (22%) | | 22 (20%) | 28 (26%) | |
| 3–4 | 294 (28%) | 11 (31%) | 22 (12%) | | 22 (19%) | 11 (10%) | |
| Unknown | 207 (20%) | 7 (19%) | 28 (15%) | 0.482 | 20 (18%) | 15 (14%) | 0.406 |

*N* = number; ECOG PS = Eastern Cooperative Oncology Group Performance Status.

Median lines of systemic therapy received was 1, ranging from 0 to 5. Within the full population 78% received systemic therapy, with 84% of LS and 74% of ES receiving at least one line of treatment. First-line systemic therapy was platinum doublet for 71%, oral etoposide for 6% and 1% received other first-line regimens. Second-line systemic therapy was delivered to 22% of the entire cohort of which 50% received platinum doublet, 17% received CAV, 12% received topotecan, 11% received irinotecan and 8% received oral etoposide. Third-line or greater was received by 59 patients (Table 2, Figure 1). Of the 7% that received non-platinum doublet for first-line systemic therapy, 23% were ECOG PS 2 and 68% were ECOG PS of 3–4.

**Table 2.** Treatment received in full SCLC population and patients treated with first-line platinum doublet who received second-line systemic therapy based on chemotherapy-free interval.

| Characteristics N (%) | Overall (*n* = 1048) | <90 Days (*n* = 36) | >90 Days (*n* = 183) | *p*-Value | <180 Days (*n* = 112) | >180 Days (*n* = 107) | *p*-Value |
|---|---|---|---|---|---|---|---|
| | | | Chemotherapy | | | | |
| First-line chemotherapy | 813 (78%) | | | | | | |
| Platinum doublet | 744 (71%) | 36 (100%) | 183 (100%) | | 112 (100%) | 107 (100%) | |
| Cisplatin doublet | 259 (25%) | 36 (100%) | 183 (100%) | | 112 (100%) | 107 (100%) | |
| Carboplatin doublet | 406 (39%) | 9 (24%) | 67 (36%) | | 35 (30%) | 41 (38%) | |
| Switch platinum doublet | 79 (8%) | 19 (50%) | 88 (47%) | | 56 (48%) | 51 (47%) | |
| Single agent etoposide | 63 (6%) | 8 (21%) | 28 (15%) | | 21 (18%) | 15 (14%) | |
| Other | 6 (1%) | | | 0.336 | | | 0.451 |
| Second-line chemotherapy | 229 (22%) | 36 (100%) | 183 (100%) | | 112 (100%) | 107 (100%) | |
| Platinum doublet | 103 (10%) | 1 (3%) | 92 (50%) | | 5 (5%) | 77 (72%) | |
| Single agent etoposide | 24 (2%) | 8 (22%) | 14 (8%) | | 16 (14%) | 6 (6%) | |
| Topotecan | 30 (3%) | 8 (22%) | 22 (12%) | | 21 (19%) | 9 (8%) | |
| Irinotecan | 30 (3%) | 8 (22%) | 20 (11%) | | 24 (21%) | 4 (4%) | |
| CAV | 38 (4%) | 7 (19%) | 31 (17%) | | 29 (26%) | 9 (8%) | |
| Other | 5 (1%) | 1 (3%) | 4 (2%) | <0.001 | 3 (3%) | 2 (2%) | <0.001 |
| Third-line chemotherapy | 59 (6%) | 8 (22%) | 50 (27%) | | 25 (22%) | 33 (31%) | |
| Platinum doublet | 13 (1%) | 1 (3%) | 12 (7%) | | 3 (3%) | 10 (9%) | |
| Single agent etoposide | 3 (<1%) | 0 | 3 (2%) | | 1 (1%) | 7 (7%) | |
| Topotecan | 11 (1%) | 1 (3%) | 10 (5%) | | 7 (6%) | 2 (2%) | |
| Irinotecan | 13 (1%) | 2 (6%) | 11 (6%) | | 5 (4%) | 8 (8%) | |
| CAV | 16 (1%) | 4 (11%) | 12 (7%) | | 8 (7%) | 8 (8%) | |
| Other | 3 (<1%) | 0 | 2 (1%) | 0.755 | 1 (1%) | 1 (1%) | 0.595 |
| | | | Radiation | | | | |
| Thoracic Radiation | 527 (50%) | 17 (47%) | 140 (77%) | | 68 (60%) | 89 (83%) | |
| Curative Intent | 207 (20%) | 2 (6%) | 65 (36%) | | 14 (13%) | 53 (50%) | |
| Palliative Intent | 320 (30%) | 15 (6%) | 75 (41%) | 0.005 | 54 (48%) | 36 (34%) | <0.001 |
| PCI | 117 (11%) | 3 (8%) | 43 (24%) | <0.028 | 12 (11%) | 34 (32%) | <0.001 |
| WBRT | 325 (31%) | 14 (39%) | 98 (54%) | 0.077 | 53 (47%) | 59 (55%) | 0.153 |
| | | | Surgery | | | | |
| Primary thoracic resection | 28 (3%) | 2 (6%) | 4 (3%) | 0.717 | 4 (4%) | 3 (3%) | 0.951 |

*N* = number; CAV = cyclophosphamide, doxorubicin and vincristine; PCI = prophylactic cranial irradiation; WBRT = whole brain radiotherapy.

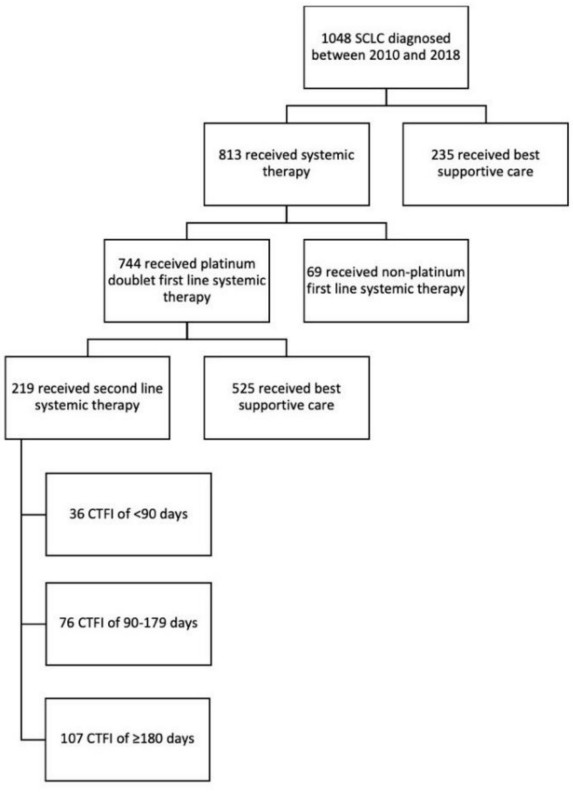

**Figure 1.** Consort diagram of small cell lung cancer dataset.

Thoracic radiation (RT) was received by 50% (*n* = 527) of the cohort with a median dose of 30 Gray. This was administered as curative intent in 40% (*n* = 207) of the population. Prophylactic cranial irradiation (PCI) was administered to 11% (*n* = 117) of SCLC patients, 84 patients with LS and 33 patients with ES. Whole brain radiation (WBRT) was received by 31% (*n* = 325) of the population of which 11 patients had previously received PCI (Table 2).

Of the 219 patients who received second-line systemic therapy after a platinum doublet, 183 patients had a CTFI of ≥90 days and 107 had a CTFI of ≥180. Patients originally treated as LS disease, received platinum doublet as second-line systemic therapy, received thoracic RT or PCI were more likely to have a CTFI of ≥90 days and ≥180 days (Table 1).

Median OS was 15.2 months in patients with LS and 5.8 months in patients with ES disease. Survival was longer in patients who received systemic therapy compared to best supportive care with LS disease (16.8 vs. 8.0 months, *p* < 0.001) and ES disease (7.8 vs. 1.3 months, *p* < 0.001) (Figure 2). The median OS for patients receiving second-line CAV was 4.6 months versus topotecan 3.8 months (*p* = 0.905) (Figure 3).

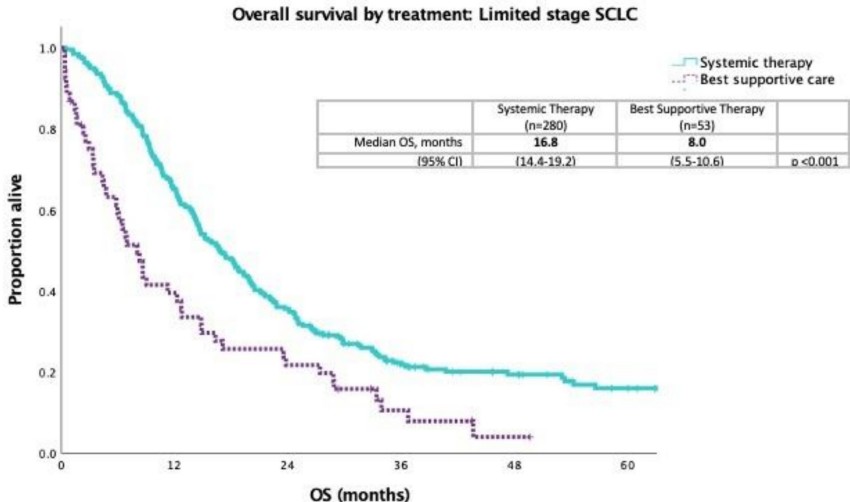

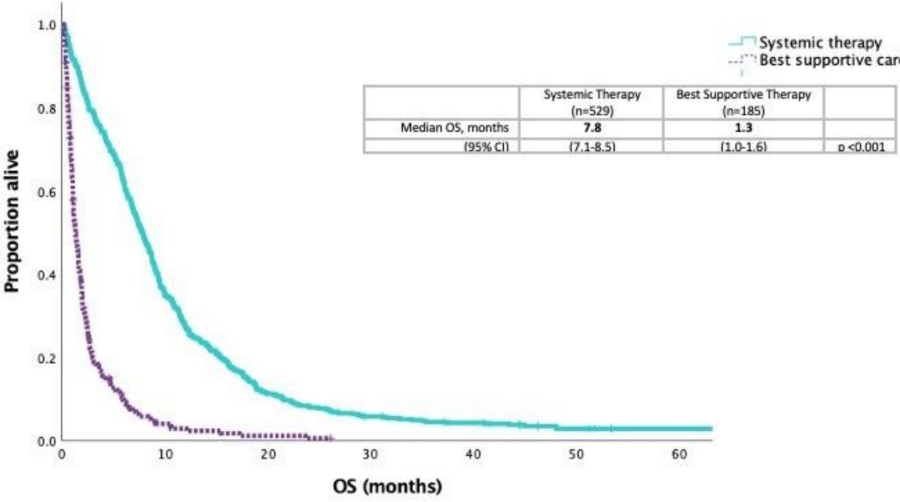

**Figure 2.** Kaplan–Meier curve for overall survival comparing patients who received systemic therapy to patients who received best supportive care in limited stage disease and extensive stage disease.

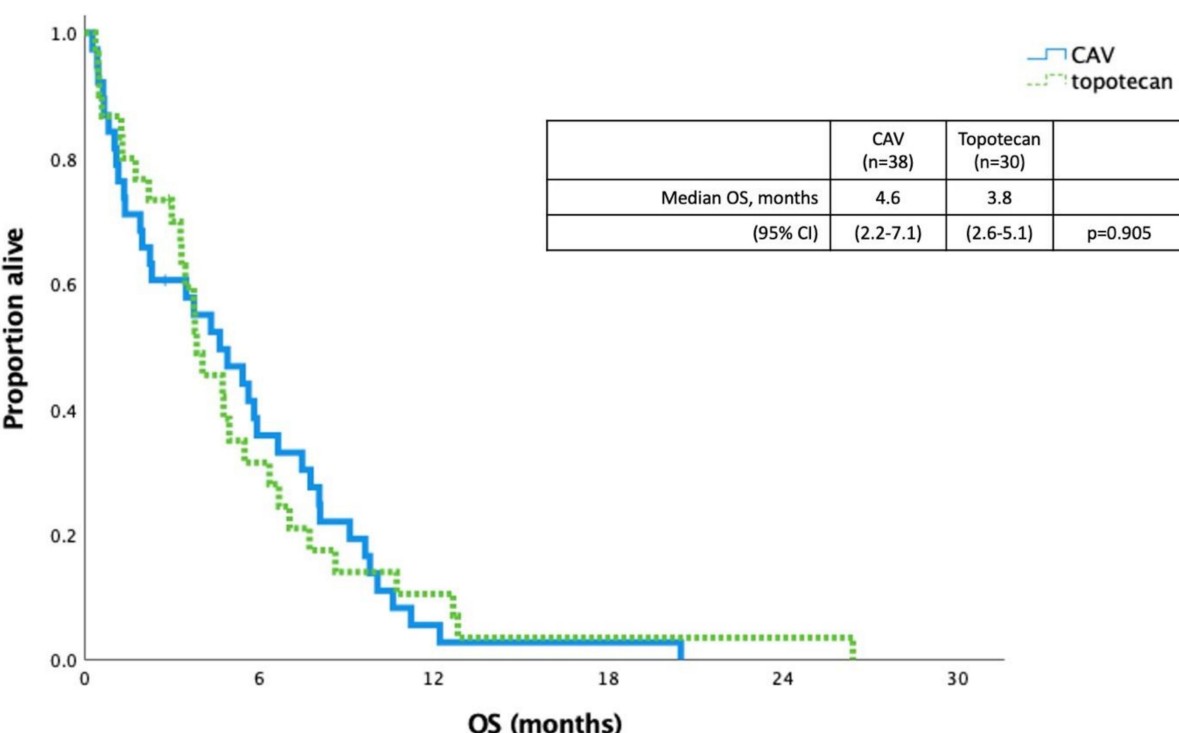

**Figure 3.** Kaplan–Meier curve for overall survival comparing patients who received second-line CAV versus topotecan.

## 4. Discussion

This retrospective review of 1048 SCLC patients highlights the challenges associated with delivering systemic therapy. In our real-world study, 29% of patients did not receive any systemic therapy despite the known high-response rates associated with chemotherapy for SCLC. Of the patients who initially received chemotherapy, only 22% went on to receive second-line treatment. Our goal of determining the eligibility of lurbinectedin shows that, despite the good tolerability, its use as a second-line SCLC treatment may be limited given patient and disease factors that hamper the use of systemic therapy.

Baseline characteristics were similar to other recently published real-world SCLC cohorts. Over the last three to four decades there has been a shift in male and female smoking habits with proportional increased smoking rates in female and decreasing rates in men [1]. Our cohort was 47% male which is a similar gender distribution to other Canadian SCLC cohorts [3,4]. Cisplatin-based chemotherapy was administered to 25% of the cohort with an additional 8% switching from one platinum agent to another. This was less than what was observed in Manitoba or Alberta with cisplatin used in the first-line setting by 74% and 43%, respectively [3,4]. Single agent etoposide was administered as first-line systemic therapy to 6% of our population, the majority of which were ECOG PS $\geq 2$, which was similar to Alberta's cohort [3]. Survival was also similar to other real-world SCLC cohorts with median OS of 16.8 months for treated LS and 7.8 months for treated ES patients [3,4].

Almost universally, LS and ES SCLC patients will have disease recurrence after completing platinum doublet [2,4,18–20]. Patients who progress during first-line platinum-based therapy are classified as platinum refractory while patients who initially respond to therapy and have disease progression within 90 days are deemed platinum resistant. Comparatively, platinum-sensitive disease is defined as disease relapse after 90 days from chemotherapy completion [11]. This distinction reflects cancer aggressiveness, with improved outcomes in patients with platinum-sensitive disease with a higher likelihood of response to second-line treatment [11]. In the current era of first-line PDL1 inhibitors with platinum chemotherapy, it remains to be seen how well these definitions will apply.

In the past, topotecan was the only FDA-approved second-line therapy. Topotecan has been evaluated against CAV after platinum therapy with ORR to topotecan of 24% that drops to 12% in the platinum-resistant population. The OS was comparable at 25.0 and 24.7 weeks, respectively [10]. In our cohort, the CAV and topotecan populations had similar median OS of 4.6 and 3.8 months, respectively, slightly lower than the study which aligns with the application of evidence in real-world populations. Topotecan has also been compared to best supportive care with a response rate of 10% in platinum-resistant and 3% in platinum-sensitive disease with median OS of 25.9 weeks [8]. While camptothecin can be shown to be active, the results have been underwhelming and evaluation of other agents has been a focus of clinical trials including lurbinectedin, a marine-derived transcription inhibitor [21].

Lurbinectedin was investigated in a phase II single arm trial, where the response rate was 45% in patients with platinum-sensitive disease and 22% in resistant disease. A post hoc analysis noted that PFS and OS was longer in patients with a CTFI of ≥90 days compared to patients with a CTFI of <90 days [13]. The promise of lurbinectedin led to ATLANTIS, a phase III study comparing lurbinectedin and doxorubicin to topotecan or CAV. The study noted a median PFS of 4.0 months for both the lurbinectedin combination and standard of care (HR 0.831), with median OS of 8.6 and 7.6 months (HR 0.967), respectively. A CTFI analysis was completed and patients with a CTFI of ≥180 days obtained PFS benefit, 8.2 versus 4.5 months, and OS was numerically improved, 12.7 versus 9.8 months. Grade 3 and 4 hematologic toxicities were less common in patients receiving lurbinectedin and doxorubicin as were treatment-related discontinuations [16]. Our study elected to examine the proportion of patients who might achieve better outcomes with lurbinectedin or lurbinectedin/doxorubicin. Using CTFI of ≥90 days as the threshold needed to obtain benefit from lurbinectedin, 17% of patients would qualify. Using CTFI ≥180 days per ATLANTIS, only 10% of patients may have qualified for second-line lurbinectedin and doxorubicin. This attrition of systemic therapy was due to declining performance status, comorbidities and patient preferences. Only a small minority of patients from our cohort would ever have been considered eligible for lurbinectedin-based therapy.

The characteristics In our cohort that predict for CTFI ≥90 or ≥180 days were LS disease at diagnosis, platinum doublet as second-line therapy, thoracic RT and PCI. The treatment variables correlated with LS disease which was not surprising as LS patients would most likely have a longer CTFI. Furthermore, a CTFI of ≥180 days may simply reflect patients with more indolent disease resulting in selection bias.

Our study is limited by the nature of this being a retrospective cohort analysis. In addition, the other known prognostic factors for SCLC, such as weight loss and laboratory values, were not routinely collected. This cohort also predates the widespread use of immune checkpoint inhibitors with platinum doublet in first-line ES SCLC, which may improve outcomes and change the rates of receiving second-line therapy. Our strengths include the real-world cohort representing a wide variety of baseline health states, geographic variation and socioeconomic status.

## 5. Conclusions

A minority of SCLC patients receive second-line systemic therapy regardless of whether the original treatment was for LS or ES disease. Only 21% of the SCLC population received second-line therapy after platinum doublet with 17% achieving a CTFI of ≥90 days and 10% achieving a CTFI of ≥180 days. Although a minority of SCLC patients receive second-line therapy, and reach a CTFI of ≥180 days, patients that fulfill these criteria may obtain a benefit from lurbinectedin. Second-line lurbinectedin may have an important role in a subset of ES SCLC patients. Unfortunately, SCLC outcomes continue to be poor and more efficacious treatments and tools to direct treatment are needed to improve outcomes.

**Author Contributions:** Conceptualization, C.H. and R.R.; methodology, C.H. and R.R.; formal analysis, C.H. and R.R.; investigation, C.H., R.R., B.L. and Z.A.-H.; data curation, C.H., R.R., B.L. and Z.A.-H.; writing—original draft preparation, C.H. and R.R.; writing—review and editing, C.H., R.R., B.L. and Z.A.-H. All authors have read and agreed to the published version of the manuscript.

**Funding:** This research did not receive any funding.

**Institutional Review Board Statement:** The study was conducted in accordance with the Declaration of Helsinki and approved by the Institutional Review Board at University of British Columbia/BC Cancer Research Ethics Board approval (H19-02381).

**Informed Consent Statement:** A waiver of consent was granted to extract and analyze data for this retrospective review.

**Data Availability Statement:** The data presented in this study are available in the above tables.

**Conflicts of Interest:** Rebekah Rittberg declares research grants from: AstraZeneca. Cheryl Ho declares research grants from: AstraZeneca, EMD Serono and Roche. Cheryl Ho declares honoraria for advisory boards from: AbbVie, Amgen, AstraZeneca, Bayer, BMS, Eisai, EMD Serono, Merck, Novartis, Pfizer, Roche, Takeda. All other authors do not have conflicts of interest to disclose.

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
