# Peer review of "Real World Patient Eligibility for Second Line Lurbinectedin Based Treatment in Small Cell Lung Cancer: Understanding Epidemiology and Estimating Health Care Utilization"

_curroncol, doi:10.3390/curroncol29120765_

Round 1

Reviewer 1 Report

The authors have retrospectively evaluated the proportion of patients with SCLC that are treated in real practice with second-line treatment and, that would be suitable for treatment with lurbinectedin based on platinum free interval. This is a relevant topic, as second-line treatment in SCLC recently includes lurbinectedin.

Comments:

1.       Please review figures in the manuscript: some data are misaligned.  

Author Response

We thank the reviewer for taking the time to provide comments on our manuscript. We have incorporated these into the manuscript.

Comments:

  1. Please review figures in the manuscript: some data are misaligned.  Thank you for this comment. We have worked to ensure that they are aligned correctly.

Reviewer 2 Report

The article entitled "Real World Patient Eligibility for Second Line Lurbinectedin Based Treatment in Small Cell Lung Cancer" presents a retrospective study on real-life patients, trying to reflect the subgroup of the population with SCLC that may benefit from the new second-line treatment with Lurbinectedin. This study collected patients with SCLC from a clinical practice database, disaggregating the clinical and epidemiological characteristics with SCLC. The study is written correctly and reflects well the characteristics of the sample, being easy to read, and the reader can quickly see how the patients with SCLC are and what treatment they have received. The authors have made a great effort in the writing and structuring of the article and that is reflected when reading the manuscript.

However, despite how well structured and completed the article is, I believe that it does not have a great value for the scientific community because it does not contribute anything new in the field of SCLC. The study focuses on the description of a series of patients with SCLC, whose characteristics are already described in the literature and are well known to all. What one expects to find when you start reading is to assess the use of Lurbinectedin, however, what the article teaches is a description of patients with SCLC independent of the use of this drug. The only noteworthy fact that is of interest to the scientific community from my point of view is that 10% of patients can benefit from the drug given the IFTC ≥ 180 days. Despite this, I believe that the finding of this data is not enough for the publication of the article.

Entering to assess technical aspects of the article, I think that the use of the language is correct and does not need modifications. The tables and figures included in the article are also correct and do not need modification. Especially the tables, show the characteristics of the sample in a simple way and with a quick glance you can make a reading of the sample. The abstract also correctly reflects the entire article, as well as the keywords. On the contrary, the title and references need a series of corrections that are set out below. The following major and minor changes are those proposed for the article: - Major changes

1. As indicated in the previous section, the study reflects the general characteristics of a database with patients with SCLC and does not bring anything new to the field of the use of Lurbinectedin, so it would be necessary to see more future studies on this drug in patients with SCLC. Descriptive studies on the population with SCLC do not contribute anything new to what is already known in oncology.

2. Title: it would be necessary to reflect what the article really is, being a review on the clinical and epidemiological characteristics of patients with SCLC.

3. Conclusions: The authors indicate that the opportunities of Lurbinectedin are scarce in SCLC, however, the results of the ATLANTIS clinical trial are promising. About 10% of patients in the authors' sample could benefit from this treatment given a ≥ 180-day IFFT. In these patients, PFS would take 4.5 to 8.2 months with the new treatment (10%). These data, although modest for being 4 months, I think are a small important advance in SCLC given the options we have today with this tumor. In view of this, I believe it would be important to change the conclusions.

4. References: only 15 citations have been used. Although these quotes about a good reflection of the literature on the subject that is being addressed, I think it is necessary to expand this point.

- Minor changes

1. Abbreviations: There are certain acronyms not specified in the text such as PCI or WBRT. It is necessary to specify all acronyms.

2. Introduction: it is indicated that the SCLC represents 13% of lung cancers. Please indicate the citation from which this data was obtained.

3. Introduction: it is indicated that 60-70% with stage ES disease have a platinum response. Not only in ES but also in LS stage. This point should be indicated.

4. Introduction: in the penultimate paragraph there is talk of a phase I combination of Doxorubicin plus Lurbinectedin. Please indicate the registration number of the clinical trial in case the reader wants to consult it.

5. Methods: indicate the country of the study where it was carried out.

6. Results: 47% of the sample were male. It is a strange fact given that most SCLCs are associated with males, explaining this fact in the discussion would be important.

7. Results: not only indicate current smoking patients, but also former ones. 

8. Results: 7% of patients did not receive platinum in the first line. Explain why these patients did not receive treatment. It is logical to think that they were unfit for platinum, however, the reason must be indicated.

9. Results: the sample shows that the OS for stage ES disease is 5.8 months. This data is lower than that of the literature, it would be interesting to explain in the discussion the possible reasons for this data.

10. Discussion: the clinical and epidemiological characteristics of the sample should be compared with what is described in the literature. Since it is a retrospective study, this is mandatory to understand if the sample is representative of the data we already know.

11. Discussion: Indicate the name or reference number of the clinical trial comparing Topotecan versus CAV.

12. Discussion Since the article revolves around the ATLANTIS trial, some assay survival curves, and toxicity data could be added to learn more clinical trial data.

13. Discussion: When survival and response data from the ATLANTIS trial are given, please provide HR or p data. 

Round 2

Reviewer 1 Report

All changes have been actioned.

Author Response

Thank you. Reviewer 1 indicated that all changes have been actioned.

Reviewer 2 Report

As indicated in the previous review, the authors' article shows us an epidemiological study on the characteristics of a sample of patients with SCLC, and how this analysis can help us better understand the possible future use of Lurbinectedin in SCLC. I believe that the authors have made a great effort to modify some points of the article which, from my point of view, did not make the subsidiary article of publication of the manuscript in the journal. The main problem one encountered when reading the article was the approach it had, its apparent content being of little interest to the scientific community. However, I think that the authors have made some very interesting changes to the article that allow it to be valued for publication. The different changes allow us to better orient the article, having a more epidemiological perspective that allows firmer conclusions to be drawn based on the results found.

The article is easy to read in general for readers and is well structured, being able to read quickly and understanding the concept you want to reach. However, I believe that the article still needs a series of major changes based on those indicated in the previous review to be subsidiary to publication that I indicate below. The rest of the minor changes listed above have all been successfully made and no further changes are needed.

1. Conclusions: in my opinion, I believe that the percentage and number of patients who can benefit from Lurbinectedin is not negligible. Therefore, I believe that a further modification of the conclusions would be necessary so that the reader can understand that their use may have a place in the future in the indication of second lines of the SCLC.

2. Introduction: in the final part of the introduction where the objective of the research is indicated, I think it would be necessary to add that it is fundamentally an epidemiological study, where through this study of the epidemiology of a population with SCLC conclusions have been drawn about the possible use of Lurbinectedin.

3. Results: it would be interesting, given that we are in a study of treatment of a second line in SCLC, to see the comparison between CAV and Topotecan in the second line. The survival curve could be included by comparing both and assessing in the discussion these survival results against those of Lurbinectedin in the second line of SCLC.

4. References: as indicated in the authors' responses, one of the references used is that of Steffens et al., I think it should be added to the references.

If the authors, make the changes correctly I think that the article could be published in the journal. I believe a great deal of effort has been made in making the changes suggested to the authors in the previous review.

Author Response

We thank the reviewer for the further comments on our manuscript. Please see our responses below.
Sincerely,
Rebekah Rittberg & Cheryl Ho

As indicated in the previous review, the authors' article shows us an epidemiological study on the characteristics of a sample of patients with SCLC, and how this analysis can help us better understand the possible future use of Lurbinectedin in SCLC. I believe that the authors have made a great effort to modify some points of the article which, from my point of view, did not make the subsidiary article of publication of the manuscript in the journal. The main problem one encountered when reading the article was the approach it had, its apparent content being of little interest to the scientific community. However, I think that the authors have made some very interesting changes to the article that allow it to be valued for publication. The different changes allow us to better orient the article, having a more epidemiological perspective that allows firmer conclusions to be drawn based on the results found.

The article is easy to read in general for readers and is well structured, being able to read quickly and understanding the concept you want to reach. However, I believe that the article still needs a series of major changes based on those indicated in the previous review to be subsidiary to publication that I indicate below. The rest of the minor changes listed above have all been successfully made and no further changes are needed.

1. Conclusions: in my opinion, I believe that the percentage and number of patients who can benefit from Lurbinectedin is not negligible. Therefore, I believe that a further modification of the conclusions would be necessary so that the reader can understand that their use may have a place in the future in the indication of second lines of the SCLC. Thank you for this comment. We have further modified the conclusion to reflect your suggestion. It now reads: “A minority of SCLC patients receive second line systemic therapy regardless of whether the original treatment was for LS or ES disease. Only 21% of the SCLC population received second line therapy after platinum doublet with 17% achieving a CTFI of ≥90 days and 10% achieving a CTFI of ≥180 days. Although a minority of SCLC patients receive second line, and reach a CTFI of ≥180 days, patients that fulfill these criteria may obtain a meaningful benefit from lurbinectedin. Second line lurbinectedin may have a important role in a subset of ES SCLC patients.  Unfortunately, SCLC outcomes continue to be poor and more efficacious treatments and tools to direct treatment are needed to improve outcomes.”

2. Introduction: in the final part of the introduction where the objective of the research is indicated, I think it would be necessary to add that it is fundamentally an epidemiological study, where through this study of the epidemiology of a population with SCLC conclusions have been drawn about the possible use of Lurbinectedin. Thank you for this comment. We have changed this paragraph to now reads: “SCLC continues to have unmet systemic therapy needs. Here we undertook a retrospective study evaluating the epidemiology of a SCLC cohort over 8 years. This cohort was then used to evaluate what proportion or which ES SCLC subset will most benefit from lurbinectedin based therapy. Based on the current published results of lurbinectedin it is not clear which ES SCLC patients will most benefit from this therapy. Using CTFI we retrospectively evaluate SCLC patients to determine the potential uptake of lurbinectedin.”

3. Results: it would be interesting, given that we are in a study of treatment of a second line in SCLC, to see the comparison between CAV and Topotecan in the second line. The survival curve could be included by comparing both and assessing in the discussion these survival results against those of Lurbinectedin in the second line of SCLC.
Thank you for the comments. The following has been added to the results and discussion section.
Results: The median OS for patients receiving second line CAV was 4.6 months versus topotecan 3.8 months (p=0.905).
Discussion: In our cohort, the CAV and topotecan populations had similar median OS of 4.6 and 3.8 months respectively, slightly lower that then study which aligns with the application of evidence in real world populations.
The figure has been added.
Figure 3. Kaplan-Meier curve for overall survival comparing patients who received second line CAV versus topotecan.

4. References: as indicated in the authors' responses, one of the references used is that of Steffens et al., I think it should be added to the references. Thank you for this suggestion. We have added this to the reference list.

If the authors, make the changes correctly I think that the article could be published in the journal. I believe a great deal of effort has been made in making the changes suggested to the authors in the previous review.
(Please disregard the attachment. Once uploaded it could not be removed)

Round 3

Reviewer 2 Report

The authors have made all the requested changes and I believe that the article is suitable for publication in the journal. I do not believe that any further changes are necessary.